# Features of Nuclear Export Signals of NS2 Protein of Influenza D Virus

**DOI:** 10.3390/v12101100

**Published:** 2020-09-29

**Authors:** Lingcai Zhao, Huizhi Xia, Jingjin Huang, Yiqing Zheng, Chang Liu, Juan Su, Jihui Ping

**Affiliations:** MOE Joint International Research Laboratory of Animal Health and Food Safety, Engineering Laboratory of Animal Immunity of Jiangsu Province, College of Veterinary Medicine, Nanjing Agricultural University, Nanjing 210095, China; 2018207020@njau.edu.cn (L.Z.); 2019807145@stu.njau.edu.cn (H.X.); huangjingjin1203@163.com (J.H.); 2019107060@njau.edu.cn (Y.Z.); changliu97@yahoo.com (C.L.); sujuan@njau.edu.cn (J.S.)

**Keywords:** influenza D viruses, nonstructural protein 2, nuclear export signals, CRM1

## Abstract

Emerging influenza D viruses (IDVs), the newest member in the genus *Orthomyxovirus* family, which can infect and transmit in multiple mammalian species as its relatives the influenza A viruses (IAVs). Additional studies of biological characteristics of IDVs are needed; here, we studied the characteristics of IDV nonstructural protein 2 (NS2), which shares the lowest homology to known influenza proteins. First, we generated reassortant viruses via reverse genetics to analyze the segment compatibility and gene interchangeability between IAVs and IDVs. Next, we investigated the locations and exact sequences of nuclear export signals (NESs) of the IDV NS2 protein. Surprisingly, three separate NES regions were found to contribute to the nuclear export of an eGFP fusion protein. Alanine scanning mutagenesis identified critical amino acid residues within each NES, and co-immunoprecipitation experiments demonstrated that their nuclear export activities depend on the CRM1-mediated pathway, particularly for the third NES (136-146aa) of IDV NS2. Interestingly, the third NES was important for the interaction of NS2 protein with CRM1. The findings in this study contribute to the understanding of IDV NS2 protein’s role during nucleocytoplasmic transport of influenza viral ribonucleoprotein complexes (vRNPs) and will also facilitate the development of novel anti-influenza drugs targeting nuclear export signals of IDV NS2 protein.

## 1. Introduction

Influenza viruses are segmented, single-stranded, negative-sense, enveloped RNA viruses belonging to the *Orthomyxoviridae* genus [1]. Four types of influenza viruses are currently recognized: the influenza A viruses (IAVs) and influenza B viruses (IBVs) contain eight gene segments, while the influenza C viruses (ICVs) and influenza D viruses (IDVs) contain seven gene segments [2]. The influenza D virus, the newest member of the *Orthomyxoviridae* genus, was first isolated from swine with respiratory symptoms in 2011 in the United States [3]. Subsequent surveys revealed that cattle, not pigs, were the natural reservoirs of these type D viruses [4,5]. There are a number of animals susceptible to IDV, including sheep, goats [6], ferrets [3], and guinea pigs [7]. Importantly, serological studies have shown that a low percentage of human sera have detectable titers of specific antibodies against the influenza D viruses [8,9].

For type A influenza viruses, infection begins with viral attachment of the hemagglutinin (HA) glycoprotein to α-2, 3- or α-2, 6-linked sialic acids on the cell surface. In contrast, the hemagglutinin-esterase-fusion (HEF) glycoprotein of IDVs attaches to 9-O-acetylated sialic acids and shares an extremely similar 3D structure to the HEF of human-infecting ICVs [10]. Infected cells produce progeny virions by assembling viral proteins and vRNPs at the plasma membrane. Viral neuraminidase (NA) or HEF esterase activity cleaves cell surface sialic acid receptors to allow nascent virions to bud and release from the infected cell membrane, thus completing the virus’s life cycle [11,12].

Influenza type D viruses are most closely related to the other seven-segmented influenza family, the influenza type C viruses. The genomic coding sequences and putative proteins identified for all segments of IDVs share only modest homology to human ICVs, approximately 29%–72% mean pairwise identity. Notably, the lowest identity match is a 184 amino acid polypeptide, encoded by a spliced form of IDV segment 7. Given its genetic similarities to IAV, this small protein is designated NS2 and is implicated in mediating the export of vRNPs from the host cell nucleus. It is this nuclear export function that led to the proposal that the NS2 protein be renamed to the nuclear export protein (NEP) [13]. In addition to being related to the nuclear export of vRNPs, it has been demonstrated that NEP contributed to the viral budding process through its interaction with a cellular ATPase [14]. Furthermore, studies have demonstrated that NEP was capable of regulating the accumulation of viral RNA species, potentially leading to a switch from viral transcription during early viral replication to promote the production of genomic vRNPs [13]. In this regard, there is substantial evidence proving that the NEP mutations that increase viral RNA replication are able to confer a significant replicative advantage during mammalian adaptation of a highly pathogenic avian influenza virus [15]. We speculate that the NS2 protein of emerging IDVs may have other features that can distinguish these viruses from classical IAVs.

Here we explored the biological characteristics of IDV NS2. First, we used reverse genetics to generate IDV NS2 recombinant viruses in a conventional A/Puerto Rico/8/1934(H1N1, PR8) backbone and then characterized these viruses; unexpectedly, we could not rescue the IDV NS2 recombinant viruses despite making several attempts. Subsequently, we analyzed the nuclear export signals and identified three nuclear export signal (NES) sequences in NS2 protein of influenza D virus. We further probed their sequences to determine the essential residues. Finally, we verified that the NES sequences mediated nuclear export via a CRM1-dependent pathway. The findings in this study contributed to understanding the role of NS2 protein during nucleocytoplasmic transport of influenza vRNPs and would facilitate the development of novel anti-influenza drugs targeting nuclear export signal of IDV NS2 protein.

## 2. Materials and Methods

### 2.1. Viruses and Cells

Human embryonic kidney (HEK293T) cells were cultured in Dulbecco’s Modified Eagle’s medium (DMEM; Gibco, Invitrogen, Carlsbad, CA, USA) containing 10% fetal bovine serum (FBS; Gibco), 0.2% NaHCO3, 100 µg/mL streptomycin, and 100 IU/mL penicillin (Gibco) at 37 °C with 5% CO_2_. Madin-Darby Canine Kidney (MDCK) cells were grown in MEM containing 10% new born calf serum. Influenza A virus, A/Puerto Rico/8/1934 (H1N1, PR8) was generated by reverse genetics and inoculated into 10-day-old specific-pathogen-free (SPF) chicken embryos for viruses propagation.

### 2.2. Plasmids Construction for Viruses

The eight gene segments of A/PR/8/1934 (PR8, H1N1) were amplified from PLLB-PR8-PB2, PLLB-PR8-PB1,PLLB-PR8-NS plasmids (kindly gifted from Dr. Earl G. Brown of the University of Ottawa, Canada) and inserted into pHH21 vector. The genes of NS and M from D/bovine/Shandong/Y127/2014 virus (Genbank Source Sequence Accession: KM015511; KM015510) were chemically synthesized by Sangon Biotech (Shanghai, China). (i) pHH21-PR8NS1-NS2, pHH21-PR8NS1-DNS2, and pHH21-PR8M_PS_-DM_ORF_ construct was generated. Modified NS segments, containing the NS1 coding sequence from PR8 and the NEP coding sequence from PR8 or D/bovine/Shandong/Y127/2014, were cloned into the pHH21 vector. Modified M segment contained a complete coding sequence of M gene segment from IDV. The modified NS1 ORFs from PR8 were followed by the porcine teschovirus-1 (PTV-1) 2A autoproteolytic cleavage site (ATNFSLLKQAGDVEENPGP) [16]. Here, the splice donor and acceptor sites of the NS1 gene were silenced by site-directed mutagenesis without affecting the amino acid sequence. In addition, the NEP ORFs were flanked by the packaging signal(ps) [17], consisting of the last 60 nucleotides from the PR8 NS segment to ensure efficient packaging into the viral particle. The plasmid pHH21-PR8M_PS_-DM_ORF_, including 247 and 240 nucleotides of PR8 M segment packaging sequences at the 3’ and 5’ ends of IDV M ORF, was constructed, and the ATG start codons right before the insert site were also mutated to TTG [18]. (ii) PR8 M1 gene and IDV M1 gene were amplified by PCR with primers containing HA-tag at the 5’ ends and cloned into pCAGGS vector via standard enzyme digestion and ligation manipulation. Sequences of primers used for plasmids construction and site-directed mutagenesis were available from the authors upon request.

### 2.3. Plasmids Construction for Protein Expression

The ORF of enhanced green fluorescent protein (eGFP) was cloned into the EcoRI-NotI sites of plasmid pCAGGS-MCS, and Flag-tag sequence was fused by PCR to the N-terminus of the eGFP ORF to facilitate immune detection. Truncated mutants corresponding to the amino acid mutations or the point mutants designated in IDV NS2 protein NES motifs were constructed via overlap PCR using the plasmid pCAGGS-eGFP as a template with corresponding primers and cloned into the protein expression vector pCAGGS. All truncated mutants and short nuclear export signals (NESs) were fused to eGFP C terminal. CRM1 gene was amplified from HeLa cell, cellular RNA was extracted by lysing of the cells with TRIzol LS reagent (Life Technologies, Inc.). The RNA was reverse-transcribed into cDNA using Moloney murine leukemia virus (M-MLV) reverse transcriptase (New England BioLabs). The segment was amplified with the Phanta Super-Fidelity DNA Polymerase (Vazyme biotech) using segment-specific primers containing HA-tag at the 5’ ends as previously described [19]. Sequences of primers used for plasmids construction and site-directed mutagenesis are available from the authors upon request.

### 2.4. Site-Directed Mutagenesis

Point mutant plasmids were constructed using a site-directed mutagenesis kit (Beyotime, China). Five hundred nanograms plasmid DNA was used for the reaction with mutagenic oligonucleotides. All point mutations were verified via Sanger sequencing.

### 2.5. Virus Rescue

Virus rescues were performed as previously described by the “twelve-plasmid” reverse genetics system [20]. Briefly, 1 µg of each four protein-expressing plasmid and 0.2 µg of each vRNAs-expressing plasmid (pHH21-PR8-PB2, pHH21-PR8-PB1, pHH21-PR8-PA, pHH21-PR8-NP, pHH21-PR8-HA, pHH21-PR8-NA, pHH21-PR8-M, and pHH21-PR8-NS or the NS chimeric constructions) were combined with 12 µL Lipofectamine 2000 (Invitrogen) (2 µL per µg DNA), and the mixture was incubated at room temperature for 30 min and then transferred to 60%–70% confluent monolayer HEK293T cells on six-well plates. After 6 h, the mixture was removed from the cells and replaced with Opti-MEM (Gibco-BRL). Forty-eight hours post-transfection, the HEK293T culture supernatants were collected and inoculated into 10-day-old SPF chicken embryos for virus propagation.

### 2.6. Immunofluorescence

HEK293T cells were seeded on glass coverslips and transfected with the eGFP-tagged fusion constructs to yield cells expressing eGFP-tagged fusion proteins. Leptomycin B (LMB) (S1726; Beyotime) was then added to the desired wells at 6 h post-transfection with a final concentration of 11 nM. Twenty-four hours post-transfection, the cells were washed with phosphate-buffered saline (PBS), fixed with 4% paraformaldehyde (pH = 7.4) for 20 min, permeabilized with 0.2% Triton X-100 in PBS for 15 min, and then stained with 4,6-diamidino-2-phenylindole (DAPI) for 10 min. For the indirect immunofluorescence assay (IFA), MDCK cells were seeded in glass-bottom plates and infected following standard procedures. Subsequent to incubation, cells were fixed with 4% paraformaldehyde and permeabilized with a solution of PBS containing 0.2% Triton X-100 for 10 min (Sigma, San Luis, MO, USA). Then, the cells were washed and stained for 1 h with primary antibodies followed by staining with fluorescence secondary antibodies in a 5% bovine serum albumin (BSA; Thermo Scientific, Waltham, MA, USA) solution in PBS. Antibodies used in this study were mouse anti-Flag Mab (F1804; Sigma-Aldrich), mouse anti-NP Mab (Cambridge biologics), rabbit anti-NS2 PcAb (H1N1), goat anti-mouse FITC-labeled secondary antibody (172–1806; KPL), and goat anti-rabbit Alexa Fluor 546-labeled secondary antibody (A21085; Invitrogen). Finally, images were acquired using confocal microscopy (Nikon, Japan) equipped with micro-objective (Plan Apo 60×/1.40, oil immersion, Nikon, Japan) and Microscope eyepiece (CFI, 10×/22, Nikon). Quantification of fluorescence signal was performed in ImageJ by counting 10–20 cells for each construct, and the ratio of IntDEN within nuclear to cytoplasmic indicates nuclear export activity of NS2 protein [21].

### 2.7. Western Blot

Infected or transfected cells were washed three times with cold PBS and lysed with cold lysis buffer (1% Triton X-100, 1 mM phenylmethanesulfonyl fluoride (PMSF) in PBS) for 30 min. Lysates were clarified by centrifugation at 12,000× *g* for 10 min at 4 °C. Proteins in the lysates were separated by SDS-PAGE electrophoresis, transferred to nitrocellulose membranes, and then probed with the antibodies indicated in the figure legends. Finally, the membranes were incubated with enhanced chemiluminescence (ECL) reagents (Vazyme, China).

### 2.8. Co-Immunoprecipitation(Co-IP)

HEK293T cells were co-transfected with Flag-NEP truncation mutants and HA-CRM1. Transient transfected cells were washed twice with PBS and lysed in NP-40 lysis buffer (50 mM Tris-HC1, pH 7.4.150 mM NaC1.1% NP-40, sodium pyrophosphate, β-glycerophosphate, sodium orthovanadate, sodium fluoride, EDTA, leupeptin) supplemented with protease inhibitor (Beyotime). Whole-cell lysate was firstly precleared with protein A/G slurry (D2117; Santa Cruz). After centrifugation at 1000× *g* for 5 min at 4 °C, supernatant was incubated with 1 μg of mouse anti-Flag MAb (F1804; Sigma-Aldrich) and 20 μL of protein A/G slurry (D2117; Santa cruz) and incubated with rotation for 4 h at 4 °C. Immunoprecipitated samples collected by centrifugation were washed with NP-40 lysis buffer four times. The final pellet was dissolved in 2× SDS loading buffer for SDS-PAGE and Western blotting. Immunoprecipitations and the whole-cell lysates were probed with mouse anti-Flag MAb (F1804; Sigma-Aldrich), rabbit anti-Flag PcAb (9121; HUABIO), and mouse anti-HA MAb (H9658; Sigma-Aldrich).

## 3. Results

### 3.1. IDV NS2 Gene Could Not Functionally Replace IAV NS2 Gene in A/PR/8/1934(H1N1) Backbone

Influenza D viruses, as a new member of the family *Orthomyxoviridae*, still have many characteristics to be elucidated, especially its NS2 protein. To investigate whether the IDV NS2 protein has other differing characteristics, we first tried to generate IDV NS2 recombinant viruses in a PR8 IAV backbone (Figure 1A). Unfortunately, we could not generate viable IDV NS2 recombinant viruses by reverse genetics in this way, even though recombinant virus was able to be rescued when PTV-1 2A sequence was inserted into PR8 NS gene (Figure 1B) [16]. In the currently proposed daisy chain model of nuclear export, M1 interacts with the vRNPs via the C-terminus of NS2 [13]. Therefore, we introduced the M segment of IDV to generate DNS2/DM recombinant viruses within PR8 backbone further. Surprisingly, we could not rescue the DNS2/DM viruses either (Figure 1A).

Previous literature indicated that complete gene segments with or without noncoding regions from H17N10 could not reassort with H1N1 human IAV [22]. Another explanation could be the incompatibility in the coding regions of the segments, which also serve to direct the incorporation of viral RNA segments into virus particles [17]. Firstly, we observed the subcellular localization of PR8 M1 and IDV NS2, as well as any possible co-localization between these two proteins. We observed that IDV NS2 protein accumulated in the cytoplasm with poor co-localization when co-transfected with PR8 M1, whereas there was a strong co-localization between PR8 M1 and PR8 NS2 (Figure 1C). This phenomenon suggested that PR8 M1 protein may not function as a bridge between IDV NS2 and vRNPs, resulting in the failure to rescue viruses [23].

### 3.2. Identification of Nuclear Export Signals (NESs) in IDV NS2 Protein

The above experimental data suggested that incompatibility and difference in function of NS2 protein between viruses of different types is one manifestation of speciation as a result of evolutionary divergence. We next explored whether IDV NS2 protein had other features that could distinguish IDVs from the other types of influenza viruses. Leucine-rich nuclear export signals of IDV NS2 protein were first predicted and analyzed using an online program (NetNES 1.1 Server) [24]. The results suggested that there were several leucine-rich amino acid sequences which could be NESs in the IDV NS2 protein. We then set out to construct IDV NS2 truncated fragments (1–30aa, 31–184aa, 1–60aa, 61–184aa, 1–90aa, 91–107aa, and 108–184aa), which were inserted into the frame of eGFP (Figure 2A). Examining subcellular localization via fluorescence microscopy, we found that amino acid sequences 31–184aa, 61–184aa, 1-90aa, 91–107aa, and 108–184aa could mediate the nuclear export of eGFP fusion protein, whereas amino acid sequences 1–30aa and 1–60aa could not (Figure 2B, Figure 2C, and Appendix A).

To determine the minimum NES motifs of the predicted amino acid regions (66–75aa, 97–107aa, and 108–184aa), we constructed eGFP-NS2 (66–75aa, 97–107aa) truncated derivatives (66–74aa, 67–75aa, 97–107aa, 97–105aa, 97–102aa, 98–107aa, 98–105aa, and 99–107aa). As for amino acid sequence 108–184aa, we speculated that NES may exist in amino acid sequence 136–150aa, which has a leucine-rich region, and we constructed truncated derivatives of this region [24]. We constructed and transiently expressed truncated segments of IDV NS2 fused to eGFP to examine subcellular localization via fluorescence microscopy. We found both eGFP-NS2_66-74aa_ and eGFP-NS2_67-75aa_ were distributed throughout the cytoplasm and nucleus losing the function of nuclear export, suggesting amino acid sequence 66–75aa was the minimum NES motif (Figure 3A and Appendix A). Similarly, we found amino acid sequence 97–107aa showed the lowest ratio of IntDEN within nuclear to cytoplasmic (Figure 3B and Appendix A). As expected, amino acid sequence 136–146aa and amino acid sequence 136–142aa demonstrated nuclear export; therefore, amino acid sequence 136–142aa may be taken as a minimum NES motif (Figure 3C and Appendix A). Taking all these results together, we found there were three NES motifs within the IDV NS2 protein.

### 3.3. Identification of Key Amino Acids in NES Motifs via Alanine Scanning Mutations in eGFP Fusion Proteins

We had determined that there were three NES motifs within IDV NS2 protein: amino acid sequences (residues 66–75: LRNQLTALRI; residues 97–107: LLLPLMRNLEM; and residues 136–146: LVSLIRLKSKL). For convenience, we defined these sequences in order as NES1: _66_LRNQLTALRI_75_, NES2: _97_LLLPLMRNLEM_107_, NES3: _136_LVSLIRLKSKL_146_. Classic nuclear export signals were reported to comprise four spaced hydrophobic residues (denoted Φ1–Φ4) and to follow the consensus Φ1-(x)2-3-Φ2-(x)2-3-Φ3-x-Φ4, where x is an amino acid that is preferentially charged, polar, or small [25]. Because NES1 is very similar to PKI NES (LALKLAGLDI) [25], which exemplifies the most common spacing of the hydrophobic positions (Φ1xxxΦ2xxΦ3xΦ4), we speculated that the critical hydrophobic residues in 66–75, highlighted in bold, denote Φ1–Φ4: **L**RNQ**L**TA**L**R**I**. We mutated the bold, predicted critical Φ1–Φ4 hydrophobic residues within amino acid sequence (97–107aa: **L**LLP**L**MRN**L**E**M**) and bold, predicted Φ1–Φ6 hydrophobic residues within amino acid sequence (136–146a: **LV**S**LI**R**L**KSK**L**). Subcellular localization of transiently expressed eGFP-NS2 fusion proteins was examined via fluorescence microscopy. We found NES1 Φ4(I75A) mutations nearly abolished nuclear export activity, while NES1 Φ1(L66A) was not critical, which was similar to prototypical classic NES (Figure 4A and Appendix A). Unexpectedly, all four NES2 mutants (Φ1: L97A, Φ2: L101A, Φ3: L105A, Φ4: M107A) could mediate nuclear export of eGFP protein (Figure 4B). Subsequently, we constructed multiple-site mutants (Φ3, 4; Φ2, 3, 4) and single mutant (Φ2L: L98A and Φ3L: L99A). We found single mutant (Φ2L: L98A) and multiple-site mutants were both distributed throughout the cytoplasm and nucleus, losing nuclear export activity and suggesting that L98A mutation is critical (Figure 4B and Appendix A). Subcellular localization of NES3 mutant Φ4 suggested that I140 residue was important, and NES3 mutant Φ2 (V137A) was also slightly retained in the nucleus (Figure 4C and Appendix A). All these data suggest that Φ2 (L70), Φ3 (L73), and Φ4 (I75) were critical residues within NES1. Single Φ residues were dispensable within NES2, but mutant Φ2L (L98) was critically important. Finally, Φ4 (I140) within NES3 played a key role in mediating nuclear export of eGFP protein.

### 3.4. Nuclear Export Activity of Three Different NESs Depend on a CRM1-Mediated Pathway

We next investigated whether the three different NESs mediated nuclear export via a CRM1-dependent pathway. CRM1 is distantly related to receptors that mediate nuclear protein import, and previous reports showed it interacted with the nuclear pore complex [13]. To examine subcellular localization and interactions of CRM1 and eGFP-IDV NS2 protein or truncated NESs fusion proteins, HEK293T cells cultured on slides were co-transfected with HA-CRM1 and eGFP-NS2 (60–90aa, 91–121aa, 108–184aa and IDV NS2), and immunofluorescence confocal microscopy was performed 24 h later. eGFP-NS2 (60–90aa, 91–121aa, and 108–184aa) fusion protein and IDV NS2 protein were observed to co-localize with CRM1 in the cytosol and accumulated adjacent to the nucleus of HEK293T cells (Figure 5A). However, whether it interacted with CRM1 needed to be determined. We next used Flag-tagged eGFP-NS2 (60–90aa, 91–121aa, and 108–184aa) fusion proteins to co-immunoprecipitate (Co-IP) HA-tagged CRM1. eGFP-NS2 (60–90aa and 108–184aa) fusion proteins could be precipitated with CRM1, but eGFP-NS2_91-121aa_ fusion protein interacted with CRM1 only slightly compared to the Flag-eGFP control (Figure 5D and Appendix A). Of course, interactions between NS2 proteins and CRM1 may also involve other nuclear pore complexes. Leptomycin B (LMB), a cytotoxin that is shown to bind to CRM1 protein, specifically inhibits the nuclear export of proteins like Rev that carry a leucine-rich NES [26]. We found eGFP-NS2 (60–90aa, 91–121aa, 108–184aa and IDV NS2) were all sensitive to LMB, suggesting NS2 (60–90aa, 91–121aa, and 108–184aa) and IDV NS2 protein may mediate the nuclear export of eGFP fusion protein via a CRM1-mediated pathway (Figure 5B,C). The above data showed that eGFP-NS2 (60–90aa, 91–121aa, 108–184aa, and IDV NS2) co-localized and interacted with CRM1 to mediate nuclear export of eGFP fusion protein.

### 3.5. The Third NES (136–146) of IDV NS2 Is Important for the Interaction of This Protein with hCRM1

Previous reports suggested that the IAV NS2-NES was not crucial for the interaction of this protein with CRM1, but crucial for the formation of the ternary export complex with Ran-GTP [27]. To determine the region that is crucial for nuclear export of IDV NS2 protein, we constructed a series of mutants based on our previously successful scanning mutagenesis: 66–75mut (**A**RNQ**A**TA**A**R**A,** hydrophobic amino acid to Alanine mutation shown in bold), 97–107mut (**AAA**P**A**MRN**A**E**A**, hydrophobic amino acid to Alanine mutation shown in bold), 136–146mut (**AA**S**AA**R**A**KSK**A**, hydrophobic amino acid to Alanine mutation shown in bold) (Figure 6A). Considering that single Φ residues are dispensable in their sequence context and that each Φ-binding pocket accepts various hydrophobic residues, the observed wide range of specifically recognized NESs are reasonable [25]. To accurately analyze the important role of individual NESs, we constructed multiple-site mutants including crucial hydrophobics and other hydrophobic residues within NES motifs. Meanwhile, to further determine whether all three NESs were essential, we constructed a series of NES-deleted versions of IDV NS2, named 66–75aaDel, 97–107aaDel, 136–146aaDel, 66–75aaDel+97–107aaDel, 97–107aaDel+136–146aaDel, 66–75aaDel+136–146aaDel, and 66–75aaDel+97–107aaDel+136–146aaDel (Figure 6A). A co-IP experiment was performed, and we found three NES-mutated versions of IDV NS2, which can interact with CRM1, suggesting single NES mutants were dispensable in their sequence context (Figure 6B and Appendix A). Interestingly, NES66–75del, NES97–107aaDel, and NES66–75aaDel+NES97–107aaDel versions of IDV NS2 could interact with CRM1, while NES136–146aaDel, NES 136–146aaDel+NES 97–107aaDel and NES 136–146aaDel+NES 66–75aaDel mutant could not (Figure 6B), suggesting that the third IDV NS2-NES(136–146aa) was important for the interaction of this protein with CRM1. Of course, NES 66–75aaDel+NES 97–107aaDel+NES 136–146aaDel mutant showed no interaction with CRM1 (Figure 6B), indicating that there were only NESs in full sequence context of IDV NS2 protein. Moreover, when the subcellular localization of these eGFP fusion proteins was examined, we found NES66–75aaDel+NES97–107aaDel+NES 136–146aaDel mutant showed punctate aggregation in the cytoplasm(Figure 6C,D). Taken together, the single NES of IDV NES2 was dispensable, but NES (136–146aa) was important for the interaction of this protein with CRM1.

### 3.6. Sequence Analysis of NS2 Protein Sequence in the Family Orthomyxoviridae: The Influenza Viruses A, B, C, and D

Lastly, we comprehensively analyzed the sequences of NS2 proteins from each influenza type: A, B, C and D. Although the NS2 sequences share low identity across the genus *Orthomyxoviridae*, members belonging to any one type of influenza were conserved at most positions, suggesting a common 3D protein structure. Sequence identity of influenza D viruses indicated that IDV was related to ICV more closely than IAV or IBV. The secondary structure elements were defined based on PDB file 1PD3 (A/Puerto Rico/8/1934 H1N1) using ESPript server online [28]. Upon aligning the sequences from all types of influenza viruses, the three different NES regions within IDV NS2 protein appeared to share this position conservation (Appendix A). In particular, two nuclear export signals were located in the N-terminal domain of NS2 of IAV and IBV, while two nuclear export signals of ICV and IDV were located in the C-terminal domain. Due to the increased identity between ICVs and IDVs, it is unknown whether there was a third nuclear export signal within ICV NS2 or not, which requires further experimentation to analyze in the future.

Preliminary reassortment experiments between IDV and ICV have been performed but failed to identify reassortant viruses. This absence of genetic exchange between viruses of different genera (types) is one manifestation of speciation as a result of evolutionary divergence, which also elucidated the phenomenon that the NS2 gene from IDV could not be rescued in influenza A PR8 backbone in this study.

## 4. Discussion

Influenza D virus was first described in 2011 from a pig with respiratory disease [3] however, recent evidence indicated that cattle were the major viral reservoir [4,5,29]. Genetic and antigenic studies suggested that IDV was common in bovines with respiratory disease and that at least two genetic and antigenically distinct clades were co-circulating. Members belonging to any of the four different types of influenza viruses can undergo genetic reassortment and thus readily exchange genetic information. However, reassortment between members of different types has never been reported. The low identity and novelty of NS2 protein of IDV prompt us to investigate if there are other features of IDV NS2 protein. NS2 protein was originally implicated in mediating the nuclear export of vRNP complexes, which were synthesized in the infected cell nucleus and were assembled into progeny virions at the cell membrane [13]. We wanted to determine whether the biological roles of IAV NEP and IDV NS2 proteins could be interchanged or not. Unfortunately, we failed to rescue recombinant virus through reverse genetics. The most likely explanation could be the incompatibility between type A and type D influenza viruses [29]. We next attempted to rescue recombinant virus by constructing M segment chimeric plasmid encoding M1 and M2 protein of influenza D virus, but the idea did not work.

We next focused on the nuclear export activity of IDV NS2 protein. Firstly, we elucidated that there were three minimum NES motifs: NES1 (66–75: LRNQLTALRI), NES2 (97–107: LLLPLMRNLEM), and NES3 (136–146: LVSLIRLKSKL). We constructed many truncated versions of IDV NS2-eGFP fusion protein to confirm nuclear export activity of three different NES (Figure 3 and Figure 4). We next analyzed the mechanism of nuclear export activity and critical amino acid residues within respective nuclear export signals further. Both subcellular localization and Co-IP experiments proved that IDV NS2 protein and eGFP-NS2 fusion protein interact with CRM1. Furthermore, nuclear export activity of these eGFP fusion proteins was sensitive to LMB, providing evidence that these NESs depend on a CRM1-mediated pathway (Figure 5). Mutational analysis of critical amino acid residues within respective nuclear export signals identified many essential amino acids, but it is still unknown whether these critical residues played an important role in the influenza D virus life cycle. More detailed studies are actively underway. We also analyzed the necessity of individual NESs for the nuclear export activity of IDV NS2 and found that the third NES was critical for the interaction with CRM1, suggesting that a particular NES region may influence the overall structure of IDV NS2 protein or contain the critical interaction sites.

There are three different NESs in IDV NS2, and their nuclear export activity depends on a CRM1-mediated pathway. Previous literature reported that an anti-influenza compound exerted its antiviral effect by means of inhibiting the nuclear export function of the viral nucleoprotein-nuclear export signal 3 (NP-NES3) domain [30]. Therefore, the critical NESs identified could also facilitate the development of novel anti-influenza drugs.

## 5. Conclusions

NS2 protein was implicated in mediating the export of vRNPs from the host cell nucleus to ensure the viral genomic segments were available for packaging into progeny virions. In this study, three novel nuclear export signals were identified in emerging influenza D viruses. Several specific hydrophobic amino acids in these three NSE motifs play key roles in mediating nuclear export of vRNPs. Nuclear export activity of IDV NS2 protein depend on CRM1-mediated pathway, and the third NES of IDV NS2 is important for the interaction of this protein with CRM1. The present data in this study provide novel insights into incompatibility and differences in protein function in the family of *Orthomyxoviridae*: the influenza viruses A, B, C, and D.

## Figures and Tables

**Figure 1 viruses-12-01100-f001:**
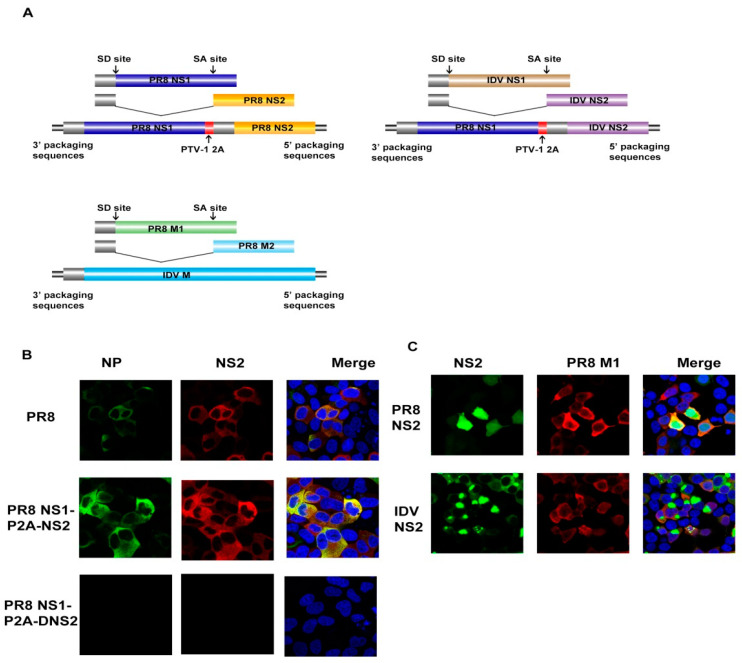
Schematic representation of the nonstructural protein chimeric gene constructs. (**A**) NS chimeric gene constructs that were used for virus rescue experiments together with the other seven gene segments from PR8. The PR8 NS1 and NS2 or influenza D virus (IDV) NS2 coding sequences are separated by the porcine teschovirus-1 (PTV-1) 2A autoprotease, and the splice acceptor site in NS was mutated to prevent mRNA splicing. The red boxes are PTV-1 2A sequences. The bottom is the structure of plasmid pHH21-PR8Mps-DMORF. (**B**) Supernatants derived from transfected HEK293T cells were used to infect MDCK cells. Cells were fixed at 24 hpi and then processed for indirect immunofluorescence assays using antibodies against NP and NS2. (**C**) HEK293T cells cultured on slides were co-transfected with eGFP-NS2 and HA-M1, and immunofluorescence confocal microscopy was performed 24 h later using anti-HA mouse monoclonal antibody followed by immunostaining with A546-labeled goat anti-mouse secondary antibody.

**Figure 2 viruses-12-01100-f002:**
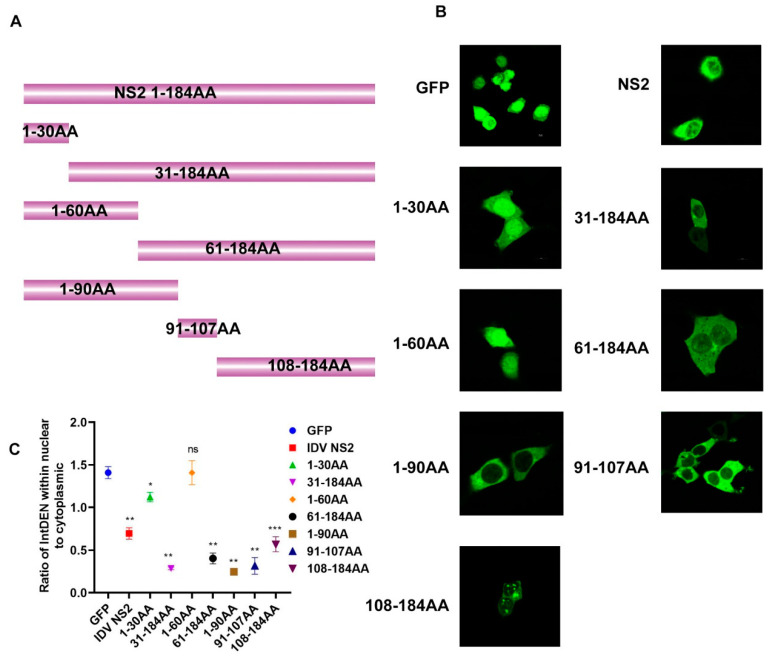
Mapping regions of nuclear export signals (NESs) of IDV NS2 protein. (**A**) Schematic representation of truncated constructs. (**B**) HEK293T cells were transfected with truncated constructs. (**C**) Quantification of fluorescence signal was performed in ImageJ. (Statistical differences are labeled according to a one-way ANOVA followed by a Dunnett s test; ns = not significant, * *p* < 0.05, ** *p* < 0.01, *** *p* < 0.001.)

**Figure 3 viruses-12-01100-f003:**
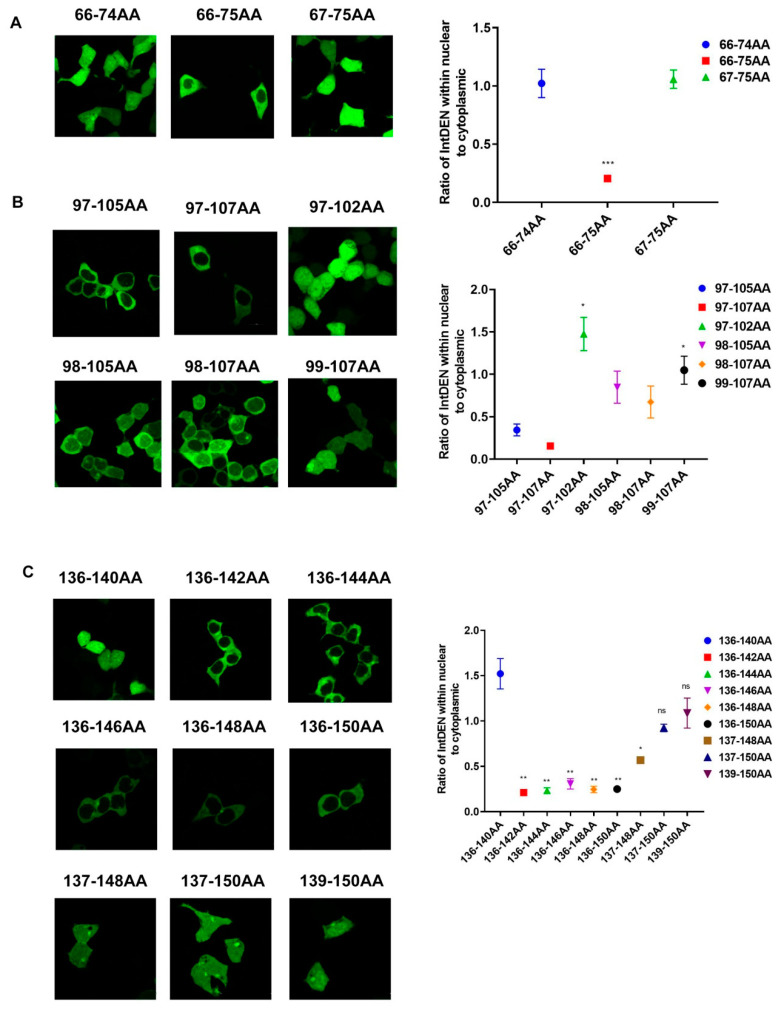
Identification of minimum NES motifs of IDV NS2 protein. (**A**) pCAGGS-eGFP-NES (66-75aa) and its truncated derivatives (66–74aa, 67–75aa) were transfected into HEK293T cells. (**B**) pCAGGS-eGFP-NES (97–107aa) and its truncated derivatives (97–107aa, 97–105aa, 97–102aa, 98–107aa, 98–105aa, and 99–107aa) were transfected into HEK293T cells. (**C**) pCAGGS-eGFP-NES (139–150aa) and its truncated derivatives (137–148aa, 137–150aa, 136–150aa, 136–148aa, 136–146aa, 136–144aa, 136–142aa, and 136–140aa) were transfected into HEK293T cells. Subcellular localization of eGFP fusion proteins were examined via fluorescence microscopy at 24 h post-transfection. Quantification of fluorescence signal was performed in ImageJ. (Statistical differences are labeled according to a one-way ANOVA followed by a Dunnett s test; ns = not significant, * *p* < 0.05, ** *p* < 0.01, *** *p* < 0.001.)

**Figure 4 viruses-12-01100-f004:**
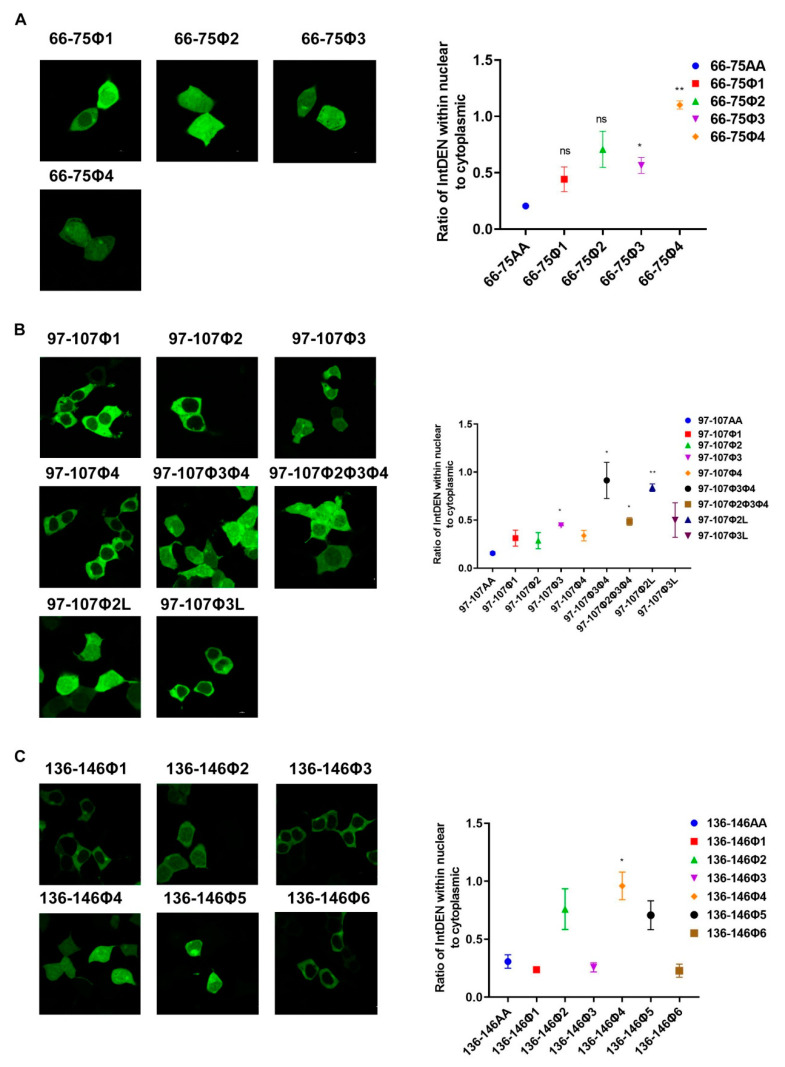
Identification of critical amino acid residues of NESs motifs mediating the nuclear export of eGFP fusion protein. (**A**) HEK293T cells growing on cover glass were transfected with pCAGGS encoding eGFP-NES (66–75aa) mutants (66–75aaΦ1, 66–75aaΦ2, 66–75aaΦ3, 66–75aaΦ4). (**B**) HEK293T cells were transfected with pCAGGS encoding eGFP-NES (97–107aa) mutants (97–107aaΦ1, 97–107aaΦ2, 97–107aaΦ3, 97–107aaΦ4, 97–107aaΦ3Φ4, 97–107aaΦ2Φ3Φ4, 97–107aaΦ3L, 97–107aaΦ2L). (**C**) HEK293T cells were transfected with pCAGGS encoding eGFP-NES (136–146aa) mutants (136–146aaΦ1, 136-146aaΦ2, 136–146aaΦ3, 136–146aaΦ4, 136–146aaΦ5, 136–146aaΦ6). Subcellular localization of eGFP fusion protein was examined via fluorescence microscopy 24 h post-transfection. Quantification of fluorescence signal was performed in ImageJ and quantification of fluorescence signal about eGFP-NES (66–75aa), eGFP-NES (97–107aa), and eGFP-NES (136–146aa) was a repeat in Figure 3. (Statistical differences are labeled according to a one-way ANOVA followed by a Dunnett s test; ns = not significant, * *p* < 0.05, ** *p* < 0.01.)

**Figure 5 viruses-12-01100-f005:**
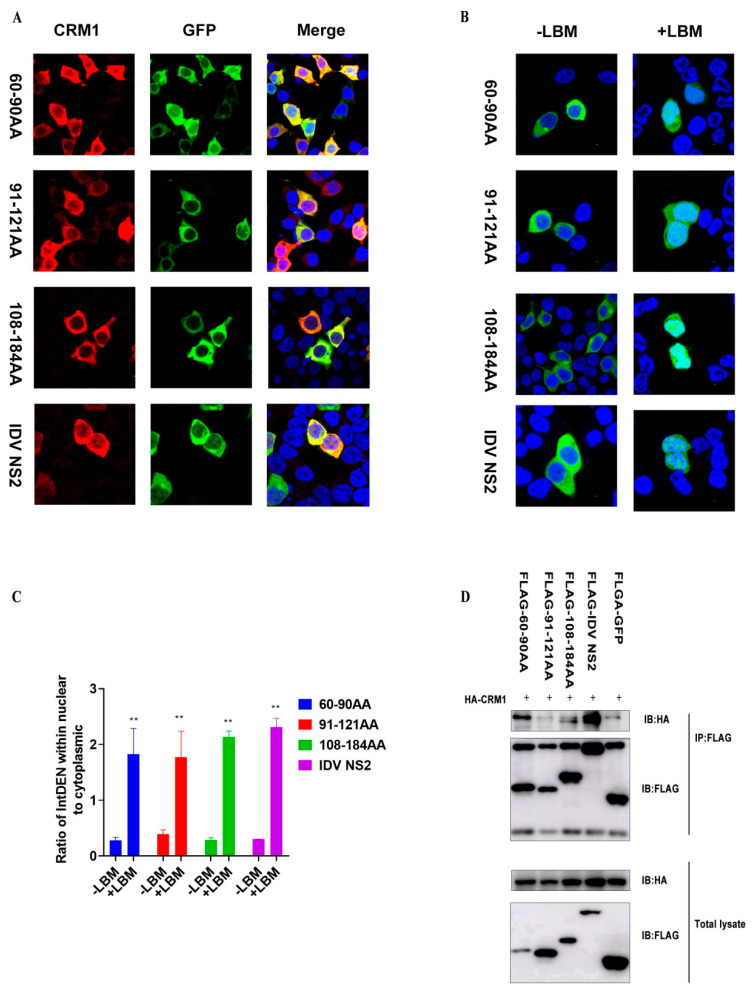
Investigation of the interaction of the three NESs of IDV NS2 with CRM1. (**A**) HEK293T cells cultured on slides were co-transfected with HA-CRM1 and eGFP-NES (60–90aa, 91–121aa, 108–184aa, and IDV NS2), and immunofluorescence confocal microscopy was performed 24 h later using anti-HA mouse monoclonal antibody followed by immunostaining with A546-labeled goat anti-mouse secondary antibody. Cells were stained with the 4, 6-diamidino-2-phenylindole (DAPI) and were examined via fluorescence microscopy. (**B**) HEK293T cells were transiently transfected with indicated plasmid pCAGGS-eGFP-NES (60–90aa), pCAGGS-eGFP-NES (91–121aa), pCAGGS-eGFP-NES (108–184aa), pCAGGS-eGFP-IDV NS2 and then treated with or without 11 nM LMB at 6 h post-transfection. Cells were then fixed, permeabilized, and stained with the 4,6-diamidino-2-phenylindole (DAPI) and were examined via fluorescence microscopy at 24 h post-transfection. (**C**) Quantification of fluorescence signal was performed in ImageJ. (Statistical differences are labeled according to a one-way ANOVA followed by a Dunnett s test; ** *p* < 0.01). (**D**) HEK293T cells were transfected with HA-CRM1 and eGFP-NES (60–90aa, 91–121aa, 108–184aa, and IDV NS2), and Co-IP was performed using mouse anti-Flag Mab and analyzed by Western blotting using mouse anti-Flag Mab and mouse anti-HA MAb.

**Figure 6 viruses-12-01100-f006:**
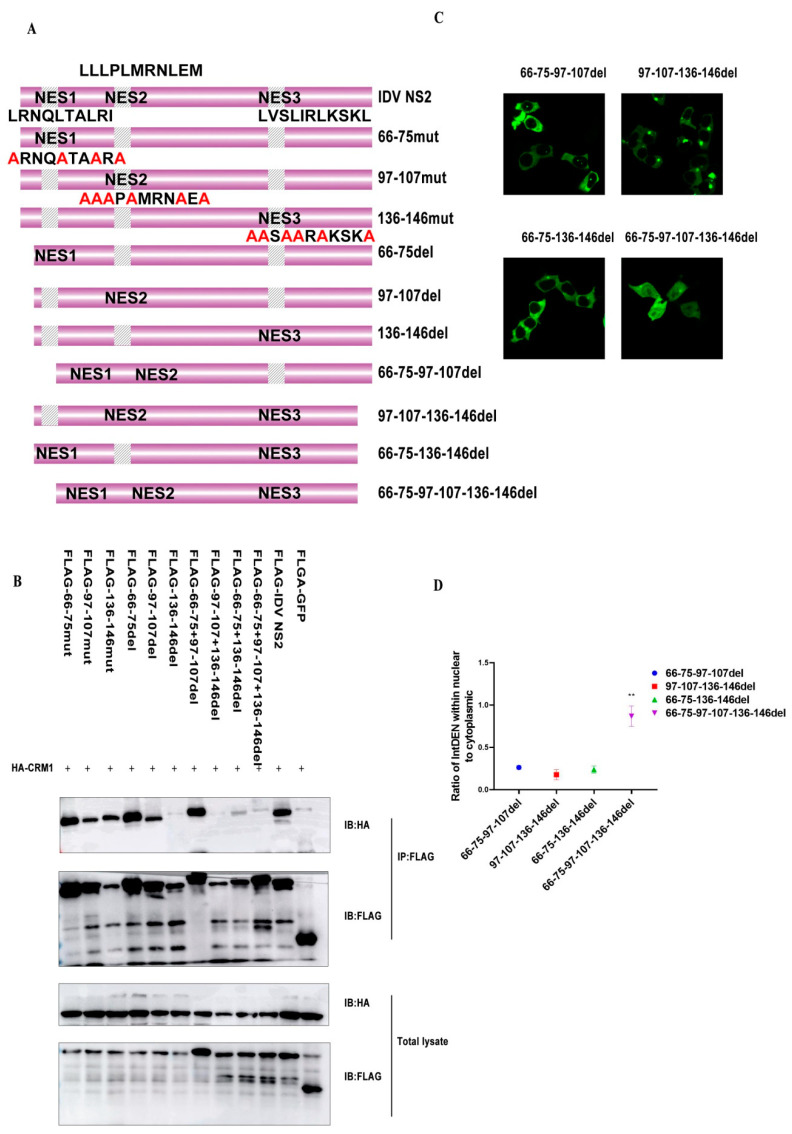
Different roles of three different NESs for the nuclear export of IDV NS2 protein. (**A**) Schematic representation of truncated constructs. (**B**) HEK293T cells were transfected with HA-CRM1 and eGFP-NES ( line 1: 66–75aaMut, line 2: 97–107aaMut, line 3: 136–146aaMut, line 4: 66–75aaDel, line 5: 97–107aaDel, line 6: 136–146aaDel, line 7: 66–75aaDel+97–107aaDel, line 8: 97–107aaDel+136–146aaDel, line 9: 66–75aaDel+136–146aaDel, line 10: 66–75aaDel+97–107aaDel+136–146aaDel, line 11: IDV NS2, and line 12: control), and then the Co-IP was performed using mouse anti-Flag Mab and analyzed by Western blotting using rabbit anti-Flag Mab and mouse anti-HA MAb. (**C**) HEK293T cells growing on cover glass were transfected with designated plasmids, and subcellular localization of eGFP fusion protein was examined via fluorescence microscopy at 24 h post-transfection. (**D**) Quantification of fluorescence signal was performed in ImageJ (Statistical differences are labeled according to a one-way ANOVA followed by a Dunnett s test; ** *p* < 0.01.)

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
