# Peer review of "Features of Nuclear Export Signals of NS2 Protein of Influenza D Virus"

_viruses, 2020, doi:10.3390/v12101100_

Round 1

Reviewer 1 Report

As previously described, the manuscript by Zhao et al describes a detailed, mechanistic of influenza D virus (IDV) NS2 protein function, particularly in regard to nuclear export, thus allowing the transport of novel viral RNPs and assembly of novel virions. The authors successfully identified 3 separate minimal IDV NS2 amino acid motifs contributing to nuclear export and show that within human cells the nuclear export process is dependent on CRM1, with the third NES site playing a critical role in this interaction. These data are confirmed elegantly using alanine-scanning mutagenesis to analyze the contribution of individual leucines within these regions at single-amino acid resolution.

The authors have adequately addressed all comments and concerns raised regarding the previous draft of this manuscript. I am happy to recommend this improved version for publication.

Author Response

Thank you very much for your comments in the multiple revisions of my manuscript.

Reviewer 2 Report

Comments for the authors of the Viruses manuscript number 945506:

The authors of the Viruses manuscript “Features about nuclear export signals of NS2 protein of influenza D virus”, present an evaluation of the NS2 protein of the newly discovered influenza D virus, with emphasis on the nuclear export properties of this protein.  The authors found 3 independent nuclear export sequences within this protein, and demonstrate that this export relies on the CRM1-mediated pathway.  This manuscript provides additional understanding related to the pathogenesis of influenza D virus, with emphasis on the nuclear export stage that is critical for spread of this virus.  While this study presents some very interesting findings, I have identified some items I would like the authors to address as they work toward improving the data presented.

General Comments:

  1. In Figure 4, it is unclear if the authors are referring to 66-74 (in the figure panels) or 66-75 (in the graph). 

Author Response

Thank you very much for your comments. We are sorry for the mistake label in the figure 4A, and the corresponding mutants are 66-75aaΦ1,66-75aaΦ2,66-75aaΦ3 and 66-75aaΦ4.

This manuscript is a resubmission of an earlier submission. The following is a list of the peer review reports and author responses from that submission.

Round 1

Reviewer 1 Report

The manuscript by Zhao et al elegantly describes a detailed, mechanistic of influenza D virus (IDV) NS2 protein function, particularly in regard to nuclear export, thus allowing the transport of novel viral RNPs and assembly of novel virions. The authors carefully identify 3 separate minimal IDV NS2 amino acid motifs contributing to nuclear export and show that within human cells the nuclear export process is dependent on CRM1, with the third NES site playing a critical role in this interaction. The use of alanine-scanning mutagenesis is a particularly impressive undertaking to analyze the contribution of individual leucines within these regions at single-amino acid resolution.

Overall the manuscript is well thought out, experiments are of a high technical quality, and the data appears well presented. For me, the attempts to construct reverse genetics viruses with PR8, while an important observation, detract slightly from the overall story presented in the rest of the manuscript. Given the strong divergence between IAV and IDV, I imagine most readers would not readily expect this to be successful therefore I’m uncertain as to why the data is featured so prominently. Perhaps the authors could consider moving part of this to SI?

Since the construction of reverse genetics viruses was unsuccessful, much of the data has been acquired through direct transfection of IDV NS2 variants and chimeras into human HEK293T cells. Since IDV infections have not yet been reported in humans, perhaps the authors could also comment on whether they expect of observations of CRM1-dependent nuclear export to be conserved across species, particularly within bovines and porcines?

The following minor points should be addressed prior to publication:

  • The authors mentioned strong differences in co-localization of PR8 M1 and either PR8 NS2 or IDV NS2 (Figure 1C) as a potential reason for unsuccessful recovery of reverse genetics viruses. They then further mention inclusion of IDV M segment (lines 172-173) with no further beneficial effect, however, data is not show to describe/support this. Did inclusion of IDV M segment improve M1-NS2 co-localization or are reasons for unsuccessful recovery more complex than this interaction?
  • Fluorescence microscopy data in Figure 6C is not referred to anywhere in the text… this data appears to correspond to the quantified data described in 6D and likely should be referred to on lines 337-338?
  • Typo in the label of Figure 6C upper right panel “97-07” should be 97-107.
  • Couple of minor grammatical errors:
    • Line 60: “evidences” should not be a plural.
    • Line 202: “constructed” use of past tense is incorrect in this sentence.

Author Response

Thank you for your comments concerning my manuscript. Those comments are very helpful for revising and improving my paper. Influenza D virus has a wide host range and a broad geographical distribution. So concerns regarding the zoonotic potential of this virus have raised. This paper is mainly about nuclear export signals of NS2 protein of influenza D virus. Previous reports suggested that influenza D viruses could replicate in several cell lines, including adenocarcinomic human alveolar basal epithelial (A549), Madin-Darby canine kidney (MDCK), Green African monkey kidney (Marc-145), human rectal tumor (HRT-18G). NS2 protein plays an important role in mediating the export of vRNPs from the host cell nucleus and CRM1-dependent nuclear export may be conserved across species. The main corrections in the paper are as following:

 In the currently proposed daisy chain model of nuclear export, M1 interacts with the vRNPs via the C-terminus of NS2. Therefore, we further introduced the M segment of IDV to generate DNS2/DM recombinant virus within PR8 backbone. But such attempts have failed and incompatibility of NS2 protein between influenza A viruse and influenza D viruse may lead to unsuccessful recovery. Of course, to further confirm the incompatibility of NS2 proteins, reverse genetics can be done in the context of type D influenza virus. But reverse genetics system of influenza D virus has not yet been established in our lab.

 To further verify that there are only three nuclear export signals in full sequence context of IDV NS2 protein, the subcellular localization of NS2 fusion proteins which contains the deletion of three nuclear export signals were examined in Figure 6C referred to on lines 337-338. As expected, NES 66-75aaDel+NES 97-107aaDel +NES 136-146aaDel mutant were distributed throughout the cytoplasm and nucleus in Fig 6C. And quantification of fluorescence signal was performed in ImageJ in Fig 6D.

We are sorry for the mistake label in the figure 6C and we have made correction. Grammatical errors in line 60 and line 202 have been corrected.

Reviewer 2 Report

Zhao et al. analyzed the nuclear export signals of the NS2 protein of influenza D virus (IDV). Through the generation of mutants and cell-based studies, they were able to identify three nuclear export signal (NES) sequences in IDV. They further probed their sequences to determine the essential residues. Finally, they performed co-IP’s and co-localization studies to show that the NES sequences mediated nuclear export via a CRMI-1 dependent pathway. While better understanding nuclear export in influenza viruses is important, there are major issues (including on experimental design, organization of text, and presentation of data) with the study as it is presented. These are further described below.

Major comments:

  1. Why was IAV chosen when making the recombinant virus? Testing influenza B and C for their compatibility would have strengthened the conclusion to this experiment.

  1. Given that the paper focuses on nuclear export, the introduction should reflect this more. There was too much detail on the viral life cycle, which was unnecessary.

  1. Methods should be placed in the methods section and not in the figure captions. For the Western Blot method section, concentrations and source of antibodies should be provided.

  1. Section 3.3. The notation is confusing. In line 240, it should state “residues 66-75” instead of 66-75aa which makes it sound like one is referring to the length of a string of amino acids. The consensus sequence given in line 244 is not in agreement with that in line 246. For the mutagenesis studies, only alanine mutations were made and tested – why not try larger hydrophobics? Additionally, other Leucine residues should have been mutated as well, to ensure that they were not important for nuclear export.

  1. Co-IPs do not prove a direct interaction. This should be made more clear. Additionally, the figures for these data are missing loading controls. Why are there up to 3 bands in the FLAG panel in figure 5D?

  1. I cannot interpret the results in section 3.6 without the data.

  1. Supplementary materials are missing.

  1. Many sentences and words need to be fixed. There are instances of incomplete sentences, e.g. lines 169-170, 177-179, 384-387, 419. Some of the sentences were confusing and I could not understand the point the authors were trying to make.

  1. There is not enough synthesis of the literature in the discussion section.

Minor comments:

  1. The title should read “Novel features of nuclear ….”. Additionally, based on the limited experimental data and comparisons to other viral NES’s, it might be a stretch calling the features discussed here “novel”.

  1. What approach was used for the experiment in lines 204-206? I presume localization via fluorescence miscroscopy, but it’s not mentioned until the next experiment is discussed.

  1. Figure 6A should have domains drawn to scale to illustrate the deletions.

Author Response

Thank you for your comments concerning my manuscript. Those comments are very helpful for revising and improving my paper. We have studied comments carefully and have made correction which we hope meet with approval. The main corrections in the paper are as following:

Influenza type D viruses are most closely related to the other seven-segmented influenza family, the influenza type C viruses. To investigate characteristics about NS2 protein of influenza D virus, we first tried to generate IDV NS2 recombinant viruses in a PR8 IAV backbone. Of course, recombinant virus may be more adaptive and more likely to be rescued if we could generate recombinant viruses by reverse genetics in influenza C virus backbone. Unfortunately, reverse genetics system of influenza C virus and influenza D virus has not yet been established in our lab. Our attempts about recombinant virus have failed and suggested that incompatibility and difference of NS2 protein between influenza A virus and influenza D virus.

This paper is mainly about nuclear export signals of NS2 protein of influenza D virus. Parts of the description about the viral life cycle have been corrected and methods about Western Blot have been corrected.

Description about NES motifs has been corrected. There are two or three residues between Φ1 and Φ2 and between Φ2 and Φ3, which exemplifies the most common spacing of the hydrophobic positions (Φ1xxxΦ2xxΦ3xΦ4).Of course, in addition to predicted critical Φ1-Φ4 hydrophobic residues, Other hydrophobic residues were tested within amino acid sequence  (97-107aa:LLLPLMRNLEM) and within amino acid sequence (136-146a:LVSLIRLKSKL), which were referred in line 248 and 249.

CRM1 is distantly related to receptors that mediate nuclear protein import and previous reports showed it interacted with the nuclear pore complex. Co-immuneoprecipitate (Co-IP) was used to investigated whether the three different NESs mediated nuclear export via a CRM1-dependent pathway. Although we can't prove a direct interaction, Leptomycin B (LMB) that is shown to bind to CRM1 protein specifically inhibits the nuclear export of eGFP-NS2 fusion protein, which proved NESs mediated nuclear export via a CRM1-dependent pathway. In figure 5D, a co-immunoprecipitation (Co-IP) experiment was performed using mouse anti-Flag antibody and lysates was determined by immunoblot analysis with mouse anti-Flag antibody and HRP-conjugated goat anti-mouse secondary antibody, so there were the heavy and light chains of immunoglobulin. However, the molecular weight of NS2 fusion protein is very close to the heavy chain of immunoglobulin, co-immunoprecipitation (Co-IP) experiment was performed using mouse anti-Flag antibody and lysates was determined by immunoblot analysis with rabbit anti-Flag antibody and HRP-conjugated goat anti- rabbit secondary antibody in figure 6B.

Considering that part 3.6 is just sequence analysis, we put it in the attachment.We can put it in the relevant part of the text if necessary. Supplementary materials have been further adjusted. Many sentences have been fixed and references are further supplemented in the discussion section. Finally, the title and parts of the description of the article are modified.

Reviewer 3 Report

Comments for the authors of the Viruses manuscript number 893546:

The authors of the Viruses manuscript “Novel features about nuclear export signals of NS2 protein of influenza D virus”, present an interesting evaluation of the NS2 protein expressed by the recently discovered influenza virus type D (influenza D virus, IDV).  Specifically, this team uses a number of molecular approaches to define the function of this protein, and identify three nuclear export regions.  This supports the assumption that this protein functions in a manner that is similar to the NS2 of influenza A virus.  The authors show this function by expressing different regions of the NS2 gene in cells individually, because they were unable to insert the IDV NS2 gene into the influenza A virus (IAV) genome.  Their plan for expressing this protein individually allows them to directly identify regions of NS2 that participate in nuclear export, including the CRM1-interacting region from 136-146aa.  Ultimately, this study will help in the development of anti-influenza drugs that target nuclear export as a way to prevent virus production by infected cells.  The fact that the IDV NS2 gene could not be inserted into the IAV genome supports the critical roles of these proteins in the virus infection cycle, and supports their classification as different virus types.  I appreciated the use of NS chimeric gene constructs (Figure 1) to really show this incompatibility, rather than attempts at simple creation of reassortant viruses by reverse genetics.  While this study presents some very interesting findings, I have identified some items I would like the authors to address as they work toward improving the data presented.

General Comments:

  1. I was wondering if the authors could describe the propagation kinetics for IDV in eggs, and maybe show a titer for the viruses used.  How well does this virus grow in eggs?  A comparison to MDCK growth would be helpful as well.  I wonder if there are any differences in the NES function between mammalian cell lines and eggs.
  2. In Figure 3 (A and B), the panel sequence did not always match the sequence in the graphs.
  3. In Figure 4, the graph showed data that were not matched with the panels.  Even if the data were redundant, it would help the reader to see it re-shown.  Maybe even a different view of cells from the same coverslip would work for these (it could be noted in the Figure legend as either a different view or a repeat of a group already shown).
  4. In Figure 6, it would help if the legends included the + symbol to read 66-75+97-107del to show that it is 2 different regions, and to match the text.
  5. I had trouble finding the supplemental data that were referred to in the manuscript.

Author Response

Thank you for your comments concerning my manuscript. Those comments are very helpful for revising and improving my paper. We have studied comments carefully and have made correction which we hope meet with approval. The main corrections in the paper are as following:

 Influenza D virus has a wide host range and a broad geographical distribution. So concerns regarding the zoonotic potential of this virus have raised. Previous reports suggested that influenza D viruses could replicate in several cell lines, including adenocarcinomic human alveolar basal epithelial (A549), Madin-Darby canine kidney (MDCK), Green African monkey kidney (Marc-145), human rectal tumor (HRT-18G). Unfortunately, influenza D viruses could not replicate in eggs and there is little research about differences in the NES function between mammalian cell lines and eggs.

 We are sorry for the confusing panels in Figure 3 (A and B) and Figure 4, the graphs have been further adjusted. The legends in Figure 6 have been corrected and supplementary materials have been further adjusted.

Round 2

Reviewer 2 Report

There are still numerous comments that have not be satisfactorily addressed. For example, there are still methods in some of the figure captions, the notation has not been consistently changed, the text has not been updated to make it more clear to readers that the co-IPs performed do not prove a direct interactions, there isn't enough synthesis of the literature, etc. The title has not been changed appropriately either and Figure 6A has not been updated based on my suggestion. 

Author Response

Thanks again for your comments. We have studied comments carefully and have made correction which we hope meet with approval. The corrections in the paper are as following:

 Firstly, we further considered the core problem of the research and the content of data presentation. We revised the title of the paper as "Identification of nuclear export signals of influenza D virus NS2 protein".

 Secondly, we have made further modifications of captions in figure 3 and figure 4. Co-IPs do not prove a direct interaction in figure 5D , we suggest that this interaction may involve other nuclear pore complexes in this paper, and subsequently we verified leptomycin B (LMB) specifically inhibits the nuclear export of eGFP-NS2 fusion protein. We redrew the scale of domains to illustrate the deletions in figure 6A. Lastly, we checked the discussion section and reorganized the literatures.